# Metastatic Progression of Osteosarcomas: A Review of Current Knowledge of Environmental versus Oncogenic Drivers

**DOI:** 10.3390/cancers14020360

**Published:** 2022-01-12

**Authors:** Guillaume Anthony Odri, Joëlle Tchicaya-Bouanga, Diane Ji Yun Yoon, Dominique Modrowski

**Affiliations:** 1INSERM UMR 1132, Biologie de l’os et du Cartilage (BIOSCAR), Lariboisière Hospital, UFR de Médecine, Faculté de Santé, University of Paris, 75010 Paris, France; joelle.tchicaya-bouanga@inserm.fr (J.T.-B.); dianejiyun.yoon@aphp.fr (D.J.Y.Y.); dominique.modrowski@inserm.fr (D.M.); 2Service de Chirurgie Orthopédique et Traumatologique, DMU Locomotion, Lariboisière Hospital, 75010 Paris, France

**Keywords:** osteosarcoma, metastasis, surgery, chemo-resistance, cancer stem cells

## Abstract

**Simple Summary:**

Osteosarcomas are heterogeneous bone tumors with complex genetic and chromosomic alterations. The numerous patients with metastatic osteosarcoma have a very poor prognosis, and only those who can have full surgical resection of the primary tumor and of all the macro metastasis can survive. Despite the recent improvements in prediction and early detection of metastasis, big efforts are still required to understand the specific mechanisms of osteosarcoma metastatic progression, in order to reveal novel therapeutic targets.

**Abstract:**

Metastases of osteosarcomas are heterogeneous. They may grow simultaneously with the primary tumor, during treatment or shortly after, or a long time after the end of the treatment. They occur mainly in lungs but also in bone and various soft tissues. They can have the same histology as the primary tumor or show a shift towards a different differentiation path. However, the metastatic capacities of osteosarcoma cells can be predicted by gene and microRNA signatures. Despite the identification of numerous metastasis-promoting/predicting factors, there is no efficient therapeutic strategy to reduce the number of patients developing a metastatic disease or to cure these metastatic patients, except surgery. Indeed, these patients are generally resistant to the classical chemo- and to immuno-therapy. Hence, the knowledge of specific mechanisms should be extended to reveal novel therapeutic approaches. Recent studies that used DNA and RNA sequencing technologies highlighted complex relations between primary and secondary tumors. The reported results also supported a hierarchical organization of the tumor cell clones, suggesting that cancer stem cells are involved. Because of their chemoresistance, their plasticity, and their ability to modulate the immune environment, the osteosarcoma stem cells could be important players in the metastatic process.

## 1. Introduction

The presence of macroscopic metastasis is the strongest predictor of survival of patients with osteosarcoma [1,2]. The mean 5-year overall survival rate for patients with metastases remains very low, below 30% [2,3,4]. The metastatic spread is heterogeneous. Indeed, metastases can be detected at the initial diagnosis or appear at the short or long term after treatment. Some patients develop metastases during treatment, which indicates refractory osteosarcoma, and these have the worst prognosis. Although the most common sites are lungs and bones [5], metastasis can appear in many other tissues. The secondary lesions can be unique or multiple, and patients with several metastases have a worse prognosis [6]. Hence, we need a better understanding of the origin of the heterogeneity and the specific mechanisms that could be targeted to stop or prevent these secondary diseases. 

Here, we review the different types of relapses/metastases that have been clinically characterized in osteosarcoma and recent knowledge of mechanisms underlying the occurrence of osteosarcoma-associated metastases.

## 2. Clinical Presentation of Metastases

### 2.1. Synchronous Metastases

Metastases that are detected at diagnosis are called synchronous because they develop in parallel with the primary tumor (Figure 1). In high-income countries, synchronous metastases are detected in 10% to 30% of patients with osteosarcoma, a proportion that has not changed in the past 50 years [2,3,5,6,7]. However, in low- and middle-income countries, with delayed diagnosis, up to 40% of patients have metastatic disease at presentation [8]. The diagnostic delay of osteosarcoma has improved in past decades in high-income countries, so the rate of patients with metastasis at diagnosis should have decreased, but, at the same time, modern imaging techniques lead to earlier detection of these metastases, possibly counterbalancing the number of patients with detected synchronous metastases [9]. We can extrapolate that we may still not be able to detect every macro and micro-metastatic disease foci, and early detection of disseminated cells might be a crucial point to better manage the metastatic disease. This is suggested by histological analyses of amputated limbs or autopsies of patients with osteosarcoma, revealing a far greater number of metastases than could be detected by imaging techniques. For example, histological analyses of resected lungs reveal more metastases than did the pre-operative CT scan [10]. Therefore, the exact proportion of patients with metastases at diagnosis could be much higher. This hypothesis is also supported by the very low proportion (only 20%) of patients who survived after being treated with surgery alone, before the introduction of chemotherapy [11]. These 20% long-term surviving patients could be truly non-metastatic at diagnosis, all the others having macro- or micro-metastases. Thus, despite improvements in imaging techniques and biopsies, we still need new tools to detect or predict metastatic disease and to discriminate metastatic and non-metastatic osteosarcoma. Identified risk factors associated with the occurrence of lung metastasis at diagnosis are delayed diagnosis, increased tumor grade and size (>5 cm), or an axial localization and presence of bone or brain metastases [7,9,12].

Synchronous metastases are mostly detected in the lungs (80% of the metastases) [13]. Synchronous bone metastases are found in 10% to 30% of the patients, and in 30% to 45% of autopsy cases [13,14,15]. Bone metastases are considered secondary events of lung lesions because most patients with bone metastases also have lung lesions. However, bone metastases can develop without or before any other non-skeletal metastases. In 1975, Enneking et al., in a systematic histological analysis of the entire limb bones after amputation for osteosarcoma, found a metastasis in the same bone in 25% of the cases. This type of lesion is called skip metastasis [16], defined as a second foci of tumor cells within the same bone as the primary lesion but separated from the primary foci by normal marrow tissue. Occasionally, skip metastasis may arise at an opposing side of a joint and is then called a trans-articular skip metastasis. Skip metastasis can be single or multiple, distal, or proximal to the primary tumor. They are found in 1.4% to 10% of patients and are best seen on MRI [17,18,19,20,21,22]. According to the American Joint Committee on Cancer staging system for primary malignant bone tumors, patients with a discontinuous tumor in the primary bone site with no regional lymph node or other distant metastasis are considered at stage 3, whereas the presence of lung metastasis is considered stage 4a, and the presence of distant bone metastases or other organ metastases is considered stage 4b.

When osteosarcoma occurs on several bones at diagnosis without lung metastases, it is called multifocal osteosarcomas (MFOSs), or multicentric osteosarcomas. This type of osteosarcoma accounts for 1% to 3% of cases [23]. Whether these MFOSs are primary osteosarcomas with bone metastases or different and independent primary osteosarcomas arising in different bones is debatable. Most of the synchronous MFOS cases have a large tumor associated with smaller tumors, which may favor a metastatic spread hypothesis, but synchronous MFOS with all lesions being the same size have also been described [24]. Histological analysis of simultaneously resected tumors from two bone sites revealed the same response rate to chemotherapy in the two lesions, which supports the hypothesis of bone-to-bone metastases from a primary tumor [25]. 

Patients with synchronous metastases have heterogeneous outcomes, depending on the number and distribution of the metastases. For example, the mean 5-year event-free survival with unilateral lung metastases is up to 75% but decreases to 0% for patients with synchronous MFOS [25]. Independent prognostic factors for patients with synchronous lung metastases are the possibility of complete metastasectomy and the response to chemotherapy [26]. Recent studies reported that 45% of patients with skip metastasis survived after 5 years, the prognosis being better with skip metastases in the same bone versus trans-articular skip metastases [3,18,19,22].

### 2.2. Metachronous Metastases

Metastases that appear after the end of treatment are considered relapses or recurrences (Figure 1). These relapses are called early when they appear during the 24 months after the end of treatment, late when they appear after 5 years, and very late after 10 years [27,28,29]. Chemotherapy has strongly reduced the early metastatic/relapse events in patients and delayed the appearance of metachronous metastases. However, more cases of late and very late recurrences up to more than 20 years have been reported. These relapses are mainly in lungs. Prior to chemotherapy, 80% to 90% of patients showed relapse, and, since the introduction of chemotherapy, this rate has decreased to 30% to 50% [6,7,14,30,31,32,33]. Furthermore, the mean time to the first relapse was about 6 to 9 months after surgery alone and is now greater than 18 months [1,13,33,34]. Chemotherapy has also changed the pattern of relapse. Most patients (80–90%) showed lung metastases, and other sites were clinically involved in less than 20% [13]. Now, 60% to 80% of patients show relapse in lungs and 20% to 40% in other sites. These sites can be bone, bowels, heart, brain, eyelid, epidural, muscle, pancreas, adrenal gland, eye, skin, stomach, or breast [14,30,32,34]. Of note, lymph node metastases seem to be rare in osteosarcoma and can be found clinically in 4% of patients but are detected in up to 28% at autopsy [13]. Chemotherapy has also changed the number of metastases at relapse, from 72% of patients with bilateral lung metastases to less than 30% [13,14,30,32,34].

In contrast, chemotherapy did not significantly change the survival of patients with metachronous metastases. In recent studies, the median post-metastasis survival is about 22 months, and the 5-year overall survival is 28% to 38% [2,5,35].

The risk factors for lung metastasis after treatment are male sex, tumor size ≥8 cm, histological grade G2, Enneking stage II, anatomic location to the distal femur, pathological subtype (conventional osteosarcoma), and high level of alkaline phosphatase at diagnosis [36]. The prognostic factors associated with better survival are the timing of metastasis [35], younger age (<60 years old), and surgical treatment of the primitive tumor [7]. The prognostic factors associated with short survival are primitive tumor size (>3 or 8 cm), histological subtype (chondroblastic versus osteoblastic), histological grade G2, Enneking stage II, early metastasis, local recurrence before metastasis, extrapulmonary metastasis, multiple pulmonary metastases, poor response to primary chemotherapy, lack of curative treatment after metastasis, including metastasectomy, increased level of alkaline phosphatase at diagnosis and lactate dehydrogenase after metastasis [5,36,37,38].

Osteosarcomas with isolated bone recurrence in the absence of any lung metastasis were previously classified as metachronous MFOSs. They represent 1% of osteosarcomas [39]. As with synchronous MFOSs, these secondary lesions could be metastatic or second independent osteosarcomas. However, it is difficult not to see a metastatic relation between the primary lesion and the secondary bone lesion, even in the absence of lung metastases, because metachronous metastases in organs other than lungs and bone (such as brain or pancreas) can be detected in the absence of lung metastases, thus showing other routes of metastatic dissemination. However, for very late metachronous MFOSs in patients with predisposing syndromes, such as p53 or Rb mutations, a second independent osteosarcoma cannot be ruled out. Patients with metachronous MFOSs have a better survival rate than those with synchronous MFOSs [23].

Local recurrence is considered when the tumor relapses at the site of the primary surgery. It accounts for 5% to 10% of relapses. Debate has been ongoing as to whether or not the local recurrences after limb salvage surgery could relate to the persistence of skip metastases in the bone that harbored the primary tumor because no local recurrence was reported after radical excision of the entire bone [17,40]. In contrast, different studies indicate a correlation between local recurrence and surgical margins, thus suggesting a possible residual disease of the primary tumor [41].

In some patients, metastases appear during neoadjuvant chemotherapy. These osteosarcomas are called refractory (Figure 1), and they are resistant to chemotherapy and have a worse prognosis of only 8% overall survival at 5 years.

### 2.3. Treatment of Metastatic Osteosarcoma

The best treatment regimen for patients with synchronous metastases is still not clear but should include chemotherapy protocols associated with primary lesion resection and metastasis resection [42]. For lung metastases, an aggressive surgical attitude including removal of metastases at several sites has been found efficient in increasing survival [4,26,43]. For patients with synchronous skip metastases, surgical resection of the primary tumor, including the skip metastases, is required [3]. Patients with synchronous metastasis in other organs or inoperable multiple metastasis often exhibit multiorgan involvement and a rapid progressive disease with a very poor short-term prognosis [3].

There is no standard treatment for patients with relapse. Patients with lung metastases should undergo metastasectomy when possible, and repeated thoracotomy can be required [42]. Complete remission is achieved only in patients with full surgical resection of all the metastasis. Second-line chemotherapy associated with surgery can increase survival as compared with surgery alone, but its exact modality is still debated [26,44,45]. In a recent review, Gazouli et al. evaluated 56 phase I and II trials for new treatment strategies in recurrent/metastatic osteosarcoma over the last 2 decades; strategies included cytotoxic chemotherapy, targeted agents, such as tyrosine kinase, and mammalian target of rapamycin inhibitors, immunotherapy, and combination approaches [44]. Most of these studies failed to show any benefit of these new drugs. High-dose chemotherapy followed by peripheral blood stem cell reinfusion showed some promising results but must be confirmed in prospective and randomized clinical trials [46]. Immunotherapy with anti-CTLA4 (pilimumab) or anti-PD1 (pembrolizumab and nivolumab) monoclonal antibodies conferred only a short stabilization of disease for a very few patients [47,48,49]. Human epidermal growth factor receptor (HER2)-directed CART cells in 16 patients with HER2-expressing osteosarcomas stabilized the disease for 12 weeks in three patients, and one patient showed 90% tumor necrosis on histological analysis after surgical excision [50].

In conclusion, the presence of metastasis during osteosarcoma development is a predictor of poor prognosis. Metastatic diseases are heterogeneous, and the timing, the number, the size, and, probably, other factors also affect patients’ outcomes. Only patients with full surgical resection of all macro metastasis can survive. However, the presence of micro metastatic foci may be frequent and very early events in osteosarcoma progression. Such cases cannot be treated by surgery and may persist after treatment because they contain refractory tumor cells that are resistant to chemotherapy. Many new therapeutic options, including targeted therapy, immunotherapy, and intensive chemotherapy with peripheral blood stem cell reinfusion, are under investigation, but none have yet shown a major change in overall survival.

## 3. Mechanisms of Metastasis Formation

### 3.1. Metastasis Heterogeneity

An important question is why consensus sets of targetable genes or miRNAs are still missing and why we cannot develop a therapeutic strategy to reduce the number of patients with a metastatic disease, despite the identification of numerous metastasis-promoting/predicting factors. The heterogenicity of the metastatic tissues could be part of the answer. Histological comparisons of primary osteosarcomas and the associated lung metastases first revealed a large variety in the dissemination process [51]. Indeed, unilateral, bilateral solitary and multiple lung lesions were described. In a group of 20 bone sarcomas that were compared to lung lesions, 60% had identical osteoblastic or chondroblastic histology, and 40% had a different histology [51]. Recently, genomic and transcriptomic analyses further documented the complex relations between primary bone tumor and metastases [52,53,54]. Wang et al. showed metastatic lesions with increased mutational burden and genome instability as compared with matched primary tumors [52]. Moreover, metastatic cell clones still have common genomic alterations with the primary tumors or, in contrast, strongly different genomic landscapes. These genetic divergences may result from different dissemination timing, with part of the metastases deriving from the mature tumor and part evolving in parallel with the bone tumor [52,53]. It was reported that metastases may be end-products of mature solid tumors or result from dissemination occurring early or continuously during tumor development [55,56]. It will be interesting to determine whether the notion of linear versus parallel evolution of the genetic pattern in metastases could overlap those of metachronous/synchronous metastasis because of the increased survival delay between bone tumor resection and metastasis initial occurrence in patients with linear evolution patterns compared to patients with parallel evolution [52]. Further genetic and transcriptomic analyses are also required to compare the different types of metastases (synchro versus metachronous and lung versus bone versus metastases in other sites).

### 3.2. Regulation of the Metastatic Capacity of Primary Tumor Cells

The formation of metastases strongly depends on the ability of tumor cells to leave the primary site. In line with this, the frequency of circulating osteosarcoma cells was found higher in patients with metastases than in those with a localized tumor, and the presence of circulating cells with mesenchymal characteristics after treatment can predict relapse and metastases occurrence [57,58]. A large number of studies focused on factors and signaling pathways that could affect the metastatic capacity of primary tumor cells. The studies analyzed cell migration/invasion using cell lines and metastasis formation in mouse models and compared transcriptomes of metastatic and non-metastatic tumors and primary tumors with metastatic tissues [54,59]. Genes that could be metastasis drivers have been extensively studied, but, more recently, epigenetic regulations by long non-coding RNAs, microRNAs (miRNAs), and circular RNAs have gained interest [60,61,62,63,64,65,66]. These non-coding RNAs also serve as prognostic markers [67,68,69]. Identifying their targets highlighted that cellular processes, such as metabolism, autophagy, and adaptation to hypoxia, are altered with the metastatic potential of tumor cells [70,71,72]. The prognosis signatures also documented modifications of the tumor environment, thus indicating that the development of metastases is associated with altered cytokine production and immune cell recruitment [71,73].

### 3.3. Epithelial–Mesenchymal Transition (EMT)

Despite being contra-intuitive given their mesenchymal origin, the EMT seems to be a key process for sarcoma cell reprogramming towards increased metastatic potential [74,75]. Recently, a prognostic signature was found to include the gene *fragile histidine triad diadenosine triphosphatase (FHIT)*, which is downregulated in osteosarcoma tissues versus normal osteoblasts. Rescuing its expression in osteosarcoma cells resulted in downregulation of N-cadherin and vimentin and upregulation of E-cadherin, which indicates that *FHIT* controls the switch between mesenchymal and epithelial markers [67]. These results demonstrate the relation between the risk of metastatic progression and EMT in patients. In addition, the presence of the mesenchymal marker N-cadherin in osteosarcoma-derived vesicles predicted metastasis progression [76]. Micro-vesicles display markers of original cells, which further links the EMT process to the metastatic potential of tumor cells. Various long non-coding RNAs and miRNAs have been shown to regulate the EMT and, thus, affect the metastatic behavior of tumor cells [74]. Especially, these RNAs modulate the expression of transcription factors, such as zinc finger E-box binding homeobox 1 (ZEB1) [77,78,79], Twist1 [80], or signal transducer and activator of transcription 3 (STAT3) [81] and signaling pathways, including the Wnt/β-catenin and NOTCH pathways [82,83].

The effectors of the Hippo pathway, YAP and TAZ, are key regulators of the plasticity of osteosarcoma cells [84]. This signaling pathway integrates mechanical and nutritional cues to drive the cell adaptation to ECM physical properties, cell junction modifications, oxygen level, or mechanical stresses (Figure 2) [85,86,87,88]. The Hippo cascade is interconnected to multiple pathways, depending on the activation of receptor tyrosine kinases, transforming growth factor β, NOTCH, Wnt, G-protein-coupled receptors, or integrins, which together modulate YAP/TAZ nuclear translocation and transcriptional activities of transcriptional enhanced associate domain (TEAD) or Smad [89,90,91]. YAP/TAZ control the EMT process directly by modulating EMT genes or indirectly by modulating signaling pathways (Figure 2) [92]. In contrast, YAP and TAZ can interact directly with the key transcription factors of the EMT, snail family transcriptional repressors (SNAIL and SLUG) and ZEB1, in mesenchymal stem cells [93]. Hence, these transcription complexes could promote the acquisition of mesenchymal/pro-metastatic capacities in undifferentiated tumor cells.

### 3.4. Role of the Microenvironment

#### 3.4.1. Extracellular Matrix

Aside from intrinsic properties, cell evasion from tumors also depends on the tumor microenvironment. Tumors are complex associations of different types of cells, including fibroblasts and endothelial and immune cells, which communicate via metabolites, cytokines, growth factors, and micro vesicles. Tumor cohesion is ensured by the extracellular matrix. Matrix proteins, such as collagens, fibronectin, and laminins, have important roles in support of cancer progression and metastases by affecting cell adhesion and migration and modulating integrin signaling and matrix metalloproteinase expression [94,95,96]. Zandueta et al. identified the matrix-Gla protein (MGP) in transcriptomic profiles of osteosarcomas. The authors showed that MGP affects endothelial adhesion, trans-endothelial migration in vitro, and tumor cell extravasation ability in vivo. Patients with metastasis had high levels of tumor and circulating MGP, and suppression of the protein resulted in reduced metastasis development in mice [97]. Hence, altering interactions between osteosarcoma cells and specific matrix proteins may offer therapeutic options to prevent the first steps of the metastasis process.

#### 3.4.2. Tumor Associated Macrophages

The role of tumor associated macrophages (TAMs) in the promotion of cancer cell invasion and metastasis capacities has been documented in diverse cancers [98]. In the same way, evidence of the contribution of these cells to the metastatic progression of osteosarcomas was reported in cell and mouse models. These studies revealed that TAMs communicate with tumor cells to activate a cyclooxygenase 2–STAT3 axis and promote the EMT [99]. Maloney et al. reported that an inhibitor of epidermal growth factor receptor acted on TAMs and reduced invasion capacity and the metastatic burden in mice [100]. Moreover, muramyl tripeptide phosphatidylethanolamine, which activates monocytes and macrophages, had encouraging effects in patients with recurrent osteosarcomas [101,102]. However, the numerous studies of TAMs in patients led to diverse conclusions about the identity of these immune cells and the association with prognosis. Mainly, TAM infiltration was associated with better prognosis [103,104]. Some authors found that higher ratios of macrophage 1 to 2 (M1/M2) better predicted overall and disease-free survival [105]. Depending on the methods, M0, M1, and/or M2 macrophages were found within tumors [103,106,107]. Zhou et al. also identified alveolar macrophages with a pro-inflammatory phenotype in lung metastatic tissues [106]. Whether macrophage-like cells could be targets to prevent or treat osteosarcoma metastasis is a major issue for improving or implementing novel immunotherapies.

### 3.5. Regulation of the Metastatic Niche

The notion of “metastatic inefficiency” was established in mouse models because of the observation that most of the tumor cells injected in the circulation died some days after metastatic site colonization [108]. Hence, metastasis formation seems to be a selective process. The primary tumors do not just provide cells with intrinsic properties to survive outside the tumors and colonize other sites; they also produce growth factors, cytokines, and miRNAs that can be transported in extracellular vesicles (EVs) to modify local and distant sites and influence metastatic niches, thereby fostering the development of disseminated tumors [109]. Macklin and colleagues isolated EVs from highly metastatic osteosarcoma cell clones and showed their preferential translocation to lungs [110]. The fusion protein Rab22a-NeoF1 and its partner PYK2 were recently found produced in EVs by subpopulations of osteosarcoma cells; they can “pre-educate” the lungs of tumor-free mice to induce a higher expression of chemokines and the recruitment of macrophages to enhance metastatic development [111]. Mechanistically, the growth factor transforming growth factor β transported in EVs induced differentiation/activation of lung fibroblasts with increased expression of fibronectin [112].

However, in other models of human tumor cell xenografts in mice, osteosarcoma cell derived EVs induced macrophage accumulation in lungs but had no effect on metastasis development [113]. This finding suggests that tumor derived EVs may not be the only required messengers. It also points out that metastasis development may require a “soil” with intrinsic permissive characteristics. Clearly, tumor-independent events affect metastatic development because metastasis outgrowth can occur in patients long after primary tumor resection. These events may be related to vascularization, metabolism, and immunity.

Thus, metastasis outgrowth is not only driven by specific gene alterations in tumor cells but also depends on the permissiveness of the environment [114,115]. The tumor-independent conditions that promote metastatic disease in osteosarcoma is a large field to be explored.

### 3.6. Metastasis Cell Founders

Two main models of tumorigenesis have evolved in parallel. The stochastic one predicts that tumor cell clones undergo Darwinian selective pressure to survive and grow. The other model includes particular tumor cells with stemness features and implies a hierarchical organization. These mechanisms are not exclusive and can be equally involved in the formation of primary osteosarcomas. Despite the extensive documentation of cancer stem-like cells in osteosarcomas, the possible contribution of these cells to the metastatic process is still unclear.

Clonal progression of osteosarcoma metastases was first supported by tracing experiments that showed a polyclonal seeding in lungs from bone tumor [116]. Of note, these experiments were conducted in immunodeficient mice and did not consider the possible selection of tumor clones by the immune system. However, phylogenetic analyses confirmed the polyclonal features of metastatic lesions in patients [52,53]. Single-cell RNA sequencing also identified diverse osteosarcoma cell populations with specialized characteristics (i.e., high proliferation, angiogenesis, interferon-α (IFN-α) and IFN-γ production, MYC expression and oxidative phosphorylation, hypoxia signaling, or inflammatory responses) [106]. By analyzing differentiation gene signatures in cell clones of chondroblastic osteosarcomas, these authors revealed that malignant chondroblastic osteosarcoma cells could be transdifferentiated into malignant osteoblastic cells. Liu and colleagues further demonstrated the presence of particular osteoblastic clones within bone tumors that are at the starting position of differentiation paths [117]. This was the first evidence in patients that osteosarcoma cell clones may not develop independently but are linked by hierarchical relationships and derive or “differentiate” from each other as shown in other types of tumors [118,119]. Together, these data further support the concept of cancer stem/precursor cells (CSCs) that contribute to osteosarcoma tumorigenesis. Different teams have described osteosarcoma cells that grow in spheres, express embryonic stem cell markers, and have high tumorigenic capacity [120,121]. We identified calpain-6 as a marker of sarcoma SCs [122] and showed that calpain-6–expressing cells are at the top of a cell hierarchy. These cells were able to rest as a slowly growing/dormant disease in lungs, months after orthotopic implantation of a few osteosarcoma cells in mice [122]. Hence, sarcoma SCs could specifically contribute to the metastatic process, but the mechanisms remain to be further elucidated.

In patients, CSCs could be selected after chemotherapy and be responsible for the residual disease owing to their chemoresistance capacity [123,124,125,126]. The most characterized CSCs are carcinoma SCs. In comparing early and advanced lung metastases of breast carcinoma, cells of low-burden metastases expressed CSC markers, whereas cells of high-burden lesions had the same gene signature as the primary tumors [127]. This finding suggests that carcinoma SCs are involved during early steps of metastasis outgrowth and that metastases could evolve similar to the primary tumors. In support of this, the acquisition of a metastatic behavior is strongly correlated with the EMT program in CSCs [128,129,130]. However, metastasis development also depends on cell plasticity and the ability to produce committed epithelial progeny. Indeed, blocking tumor cells in the mesenchymal phenotype was sufficient to inhibit metastasis progression in different models of epithelial cancer [127,131]. In the same way, the EMT program seems closely associated with both stemness and metastasis capacities during osteosarcoma progression [132,133]. On the other hand, metastatic capacity is correlated with SC features. For example, the metastatic potential of osteosarcoma cells was found correlated with aldehyde dehydrogenase 1A1 activity, a well-known CSC marker [134]. Enhanced transglutaminase 2 expression in metastatic osteosarcomas contributes to the migratory and invasive properties of tumor cells but also to stem cell phenotype [135]. Zhang et al. found that osteosarcoma SCs can undergo myofibroblastic reprogramming that mimics a process of non-tumor myofibroblasts by inducing fibrosis in lungs, which, in turn, promotes metastasis growth [136]. Hence, osteosarcoma SCs have specific features, such as the EMT, high migration capacity, chemoresistance, and plasticity, so they are strong candidates to drive an efficient metastatic process.

One possible mechanism to explain the ability to re-initiate tumor development after treatment is that CSCs modulate immune environment and can induce local immunosuppression [137]. An ancient idea is that, because cancer cells differ from normal cells owing to genetic instability and a high rate of mutations, tumor growth should induce innate and adaptive immune responses. In support of this, cancer-specific antigens have been identified, and an immune surveillance was shown to keep metastasis cells quiescent [138]. However, CSCs can overcome this constraint by intrinsic mechanisms and by modeling a specific microenvironment. For example, similar to normal SCs, CSCs express low levels of class I major histocompatibility complex [139]. In contrast, CSCs express high levels of programmed death ligand 1, a checkpoint inhibitor that suppresses T-cell activation and promotes T regulatory cell differentiation [140,141]. Alternatively, a CSC-specific microenvironment called a niche was found to comprise matrix proteins, such as tenascins, which inhibit CD4+ and CD8+ T-cell activation and proliferation and production of IFN-γ [142]. Finally, much evidence indicates that the dialogue between CSCs and the environment results in the production of a specific set of inflammatory mediators that, in turn, regulate the recruitment and differentiation of the immune cells [137]. In conclusion, CSCs are able to affect the immunity set point, that is, the balance determining pro- or anti-tumor immunity, and could thereby escape immune control.

## 4. Conclusions and Perspectives

Chemotherapy has improved the outcome of patients with a local bone tumor. However its impact on metastatic diseases is less clear. Chemotherapy has changed the timing, the pattern, and the number of metastases, but it did not significantly change the survival of patients. Only 20 to 30% of the patients with synchronous metastases and 30–40% of the patients with metachronous metastases can be cured, and, for this, full surgical resection of the metastases must be achieved. Targeted therapy, immunotherapy and intensive chemotherapy with peripheral blood stem cell reinfusion are under investigation and have not yet shown a major change in the overall survival [143,144,145,146].

Better understanding of the mechanisms of metastasis formation is, therefore, mandatory to find new therapeutic targets. In the recent years, innovative technologies have highlighted several key pathways that regulate the metastatic capacity of the primary tumor cells, such as epigenetic regulations by small and long non-coding RNAs and genomic alterations, the importance of the Hippo pathway in the EMT of osteosarcoma cells, the role of extracellular matrix and cells infiltrating the tumor in the promotion of cell mobility and metastatic capacity, and the host tissue modifications into a metastatic niche by extracellular vesicles from the primary tumor (Figure 3). Osteosarcoma stem cells specific features have been better characterized, and these are strong candidates to drive an efficient metastatic process through the modification of the immune environment.

Systematic tumor molecular profiling by DNA and RNA next generation sequencing on primary tumors and metastases biopsies, recognition of the immune molecular signatures, and identification of specific osteosarcoma stem cells molecular targets will hopefully allow the development of novel targeted therapies and may offer personalized treatments for improved clinical result in patients with metastatic disease.

## Figures and Tables

**Figure 1 cancers-14-00360-f001:**
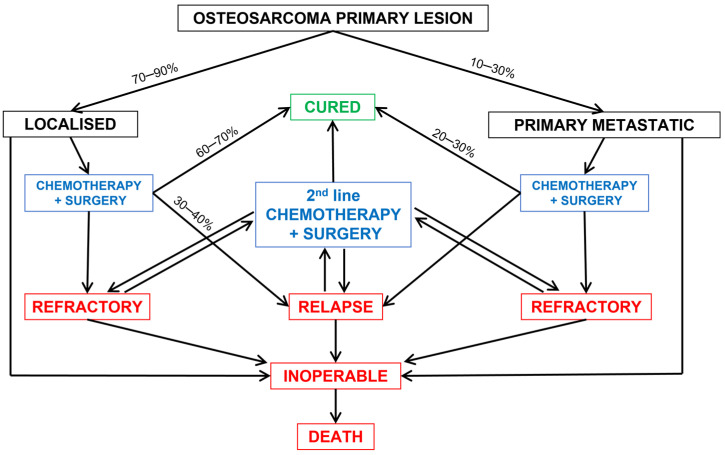
Metastatic evolution in osteosarcoma. The outcome depends on the occurrence of metastasis before diagnosis and/or relapse after treatment, the response to chemotherapy, and the possibility for full surgical resection. The text in black is for initial presentation, the text in blue is for therapeutic option, the text in red is for unfavorable evolution and the text in green is for favorable evolution.

**Figure 2 cancers-14-00360-f002:**
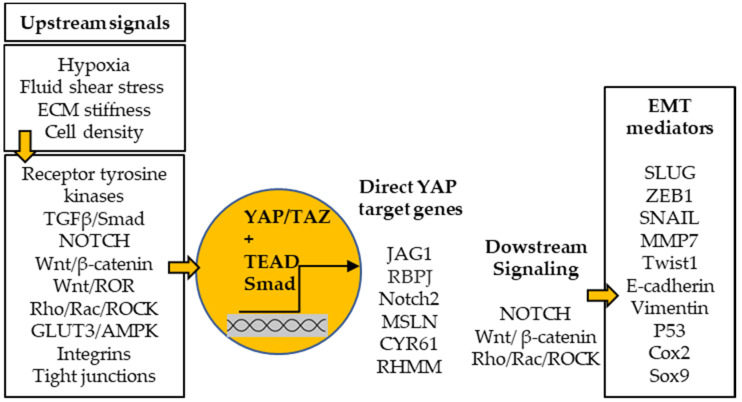
Effectors of the Hippo pathway, YAP/TAZ, are at the core of the epithelial–mesenchymal transition (EMT) regulation. Their nuclear translocation and transcriptional activities depend on microenvironment cues and the multiple signaling pathways that integrate these signals to induce cell adaptation. YAP/TAZ, in turn, modulate the expression of genes and activity of signaling pathways involved in the EMT.

**Figure 3 cancers-14-00360-f003:**
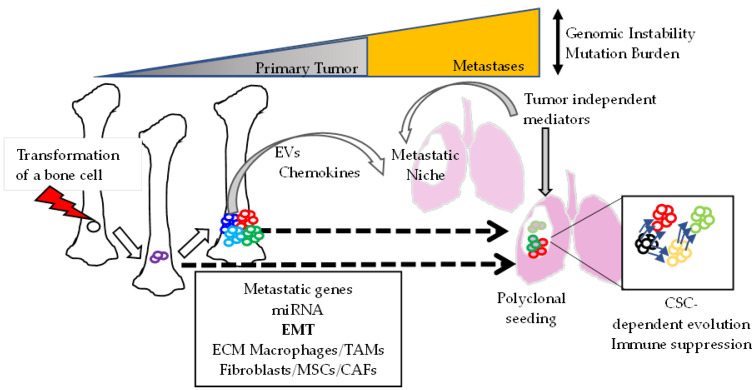
Schematic view of the formation of osteosarcoma-derived lung metastases accompanied by increased genomic alterations. The metastatic capacity of bone tumor cells depends on gene alterations (metastatic genes) and epigenetic regulation by microRNA (miRNA). Both can contribute to the epithelial–mesenchymal transition (EMT). The extracellular matrix (ECM) and tumor-associated fibroblasts (CAFs) and macrophages (TAMs) are responsible for microenvironment-mediated promotion of tumor progression. Cells can escape the primary site during early or late stages of tumor development (as indicated by the dotted arrows). The primary tumor produces factors that can travel in extracellular vesicles (EVs) to induce or activate metastatic niches. The metastatic niche could be regulated by tumor-independent factors that induce the host to be highly permissive for metastasis development. Polyclonal seeding can be associated with a hierarchical progression, with cancer stem cells (CSCs) as an initiating population of metastases. MSCs, mesenchymal stem cells.

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
