# Peer review of "Metastatic Progression of Osteosarcomas: A Review of Current Knowledge of Environmental versus Oncogenic Drivers"

_cancers, 2022, doi:10.3390/cancers14020360_

Round 1

Reviewer 1 Report

The manuscript is well written, presenting a good spectrum of information on OS metastasis.

1)  Double check on references; at least one irrelevant reference (#35) is noticed.

2) “One possible mechanism to explain chemo-resistance and the ability to re-initiate tumor development after treatment is that CSCs induce immunosuppression”. This proposal does not seem to be scientifically sound, despite the fact that CSCs can acquire various properties to evade immunity, and antitumor immunity may play an important role in tumor eradication.

Author Response

Dear reviewers, thank you for your review and your comments. Please find bellow the response to these comments. 

Reviewer 1 :

The manuscript is well written, presenting a good spectrum of information on OS metastasis.

  • Double check on references; at least one irrelevant reference (#35) is noticed.

The erroneous reference #35 has been changed and is now reference #50. This was du to a confusion between Ahmed G. et al (2019) and Ahmed N. et al (2015). All the other references have been crosschecked and no other mistake has been found.

  • “One possible mechanism to explain chemo-resistance and the ability to re-initiate tumor development after treatment is that CSCs induce immunosuppression”. This proposal does not seem to be scientifically sound, despite the fact that CSCs can acquire various properties to evade immunity, and antitumor immunity may play an important role in tumor eradication.

The sentence has been changed to “One possible mechanism to explain the ability to re-initiate tumor development after treatment is that CSCs modulate immune environment and can induce local immunosuppression”. Therefore, chemo-resistance has been removed.

Reviewer 2 :

The authors provide an interesting review of the metastatic potential of osteosarcoma and elaborate on some potential pathways involved in the process of metastases development. I would recommend few changes and improvement to increase the quality of the manuscript:

  1. Part 4.2 Metastasis cell founders - it fits better to part 3 of the manuscript. Or could be split between chapter 3 and 4.

Although we understand this point, we decided to keep the Cell founders as a single paragraphe instead of splitting it in § 3 and 4. The purpose is to stress on the heterogeneity of the metastases and to present CSC as a possible key of this problem of heterogeneity.  

  1. Chapter 4 about the heterogeneity of metastases is missing the comparison of genetic and transcriptomic profiles of primary tumors and metastases to different metastatic sights (especially pulmonary vs non-pulmonary) what could explain different rates of mets in different organs. Are there any available data on this issue?

To our knowledge there is no transcriptional or genomic data to compare metastases in different locations.

  1. Manuscript is lacking a more extensive summary and putting the results in the context of future research and clinical applications. 

The conclusion has been modified accordingly and has been renamed conclusion and perspectives.

Minor:

  1. Gene names should be written in italics : the changes have been made accordingly.
  2. Chapter 2 name should be rather "clinical presentation" than "Clinical description". The modification has been made.
  3. Figure 1 - I suggest to change "isolated" to "localised". A word isolated is rather used for isolated metastasis (single one) and for only primary tumor "localised" is more common. The modification has been done accordingly.

Reviewer 2 Report

The authors provide an interesting review of the metastatic potential of osteosarcoma and elaborate on some potential pathways involved in the process of metastases development. I would recommend few changes and improvement to increase the quality of the manuscript:

1. Part 4.2 Metastasis cell founders - it fits better to part 3 of the manuscript. Or could be split between chapter 3 and 4
2. Chapter 4 about the heterogeneity of metastases is missing the comparison of genetic and transcriptomic profiles of primary tumors and metastases to different metastatic sights (especially pulmonary vs non-pulmonary) what could explain different rates of mets in different organs. Are there any available data on this issue?
3. Manuscript is lacking a more extensive summary and putting the results in the context of future research and clinical applications. 

Minor:
1. Gene names should be written in italics
2. Chapter 2 name should be rather "clinical presentation" than "Clinical description"
3. Figure 1 - I suggest to change "isolated" to "localised". A word isolated is rather used for isolated metastasis (single one) and for only primary tumor "localised" is more common

Author Response

Dear reviewer, thank you for your comments. Please find below the response. 

Best regards. 

Reviewer 2 :

The authors provide an interesting review of the metastatic potential of osteosarcoma and elaborate on some potential pathways involved in the process of metastases development. I would recommend few changes and improvement to increase the quality of the manuscript:

  1. Part 4.2 Metastasis cell founders - it fits better to part 3 of the manuscript. Or could be split between chapter 3 and 4.

Although we understand this point, we decided to keep the Cell founders as a single paragraphe instead of splitting it in § 3 and 4. The purpose is to stress on the heterogeneity of the metastases and to present CSC as a possible key of this problem of heterogeneity.  

  1. Chapter 4 about the heterogeneity of metastases is missing the comparison of genetic and transcriptomic profiles of primary tumors and metastases to different metastatic sights (especially pulmonary vs non-pulmonary) what could explain different rates of mets in different organs. Are there any available data on this issue?

To our knowledge there is no transcriptional or genomic data to compare metastases in different locations.

  1. Manuscript is lacking a more extensive summary and putting the results in the context of future research and clinical applications. 

The conclusion has been modified accordingly and has been renamed conclusion and perspectives.

Minor:

  1. Gene names should be written in italics : the changes have been made accordingly.
  2. Chapter 2 name should be rather "clinical presentation" than "Clinical description". The modification has been made.
  3. Figure 1 - I suggest to change "isolated" to "localised". A word isolated is rather used for isolated metastasis (single one) and for only primary tumor "localised" is more common. The modification has been done accordingly.

Round 2

Reviewer 2 Report

I do not agree with the response to major comments 1 and 2.

Ad. 1 - the part about Metastasis cell founders, mainly CSCs does not explain heterogeneity but relays to why and how metastases are created thus keeping it in part 4 is inappropriate. It can be kept as a separate part but under heading 3 "Mechanisms of metastasis formation"

Ad 2 - If there is no transcriptional or genomic data to compare metastases in different locations, such information should be also included in the manuscript. Moreover, there are multiple studies evaluating differences between metastatic and nonmetastatic osteosarcoma or between primary tumor and metastatic lesion in terms of genetic and molecular parameters, such as differentially expressed genes or proteins. This issue is still not discussed in the manuscript even though it may play important role in understanding the metastatic potential of osteosarcoma.

Minor - new part about perspectives lack proper citation. The Authors state there some trials with stem cells transplant are ongoing but do not provide references. Please add a citation to the published paper, conference abstract, or NCT number of the trial.

Author Response

Dear Reviewer, thank you for your comments. Please find beneath the response. 

Best regards. 

Ad. 1 - the part about Metastasis cell founders, mainly CSCs does not explain heterogeneity but relays to why and how metastases are created thus keeping it in part 4 is inappropriate. It can be kept as a separate part but under heading 3 "Mechanisms of metastasis formation"

The organization of the manuscript has been changed accordingly: the § “heterogeneity of the metastases” became § 3.1 and was included at the beginning of the mechanisms of metastasis formation. The text concerning “Cell founders” became §3.6.

Ad 2 - If there is no transcriptional or genomic data to compare metastases in different locations, such information should be also included in the manuscript.

We added “Further genetic and transcriptomic analyses are also required to compare the different types of metastasis (synchro vs metachronous and lung vs bone vs metastases in other sites)” at the end of § 3.1 and “Systematic tumor molecular profiling by DNA and RNA next generation sequencing on primary tumors and metastases biopsies, recognition of the immune molecular signatures and identification of specific osteosarcoma stem cells molecular targets will hopefully allow the development of novel targeted therapies” in the conclusions.

Moreover, there are multiple studies evaluating differences between metastatic and nonmetastatic osteosarcoma or between primary tumor and metastatic lesion in terms of genetic and molecular parameters, such as differentially expressed genes or proteins. This issue is still not discussed in the manuscript even though it may play important role in understanding the metastatic potential of osteosarcoma.

The § 3.1 include a discussion based on comparisons of histology, transcriptomic and genomic characteristics of primary and metastatic lesions or metastatic vs non-metastatic bone tumors. We try to show how these studies further document the complex relationship between primary bone tumor and metastases.

Minor - new part about perspectives lack proper citation. The Authors state there some trials with stem cells transplant are ongoing but do not provide references. Please add a citation to the published paper, conference abstract, or NCT number of the trial.

The trials with the stem cells transplants consist of peripheral blood stem cell reinfusion after intensive chemotherapy. The descriptions have been changed in the text accordingly and several references have been added in the conclusion for the recent targeted therapy trials.

We thank the reviewers for their help to improve the manuscript

Round 3

Reviewer 2 Report

No comments